# UBFFM at the GermEval-2025 LLMs4Subjects Task:
# What if we take "You are an expert in subject indexing" seriously?

## Clara Wan Ching Ho

Universitätsbibliothek Johann Christian Senckenberg, Goethe-Universität Frankfurt
`c.ho@ub.uni-frankfurt.de`

## Abstract

This paper presents two contributions in subject classification and subject indexing of the UBFFM team at the GermEval shared task LLMs4Subjects. In Subtask 1, a fine-tune multilingual classifier is developed to assign LinSearch subject domains, achieving consistent performance across record types. For Subtask 2, an innovative generative approach is introduced by prompting to produce GND-like subject labels enriched with metadata. The pseudo-subjects are mapped to official GND terms via embedding-based similarity matching.

## 1 Introduction

Subject indexing and classification are key to organizing knowledge in digital libraries, enabling effective information retrieval and access. As scholarly content increases, automated methods have become crucial for scalable metadata generation.

Subject classification assigns documents to broad domains. LinSearch, developed by the Leibniz Information Centre for Science and Technology and University Library (TIB), categorizes publications into 28 scientific fields using text and metadata features (Technische Informationsbibliothek (TIB), 2023a), serving as a benchmark for automated classification.

Subject indexing uses specific descriptors to capture detailed content. The Gemeinsame Normdatei (GND) provides a standardized vocabulary widely used in German-speaking academia (Deutsche Nationalbibliothek, 2023). While indexing was traditionally manual, advances since the past decade have started exploring automatic approaches such as statistical machine learning, and more recently, using large language models (LLMs) for this complex task.

An initial iteration of LLMs4Subjects was previously conducted at SemEval 2025 (D'Souza et al., 2025a), where subject indexing using the GND served as the sole task. At the first phase, various LLM-based subject indexing methods were proposed and evaluated by the participating teams. The Annif team (Suominen et al., 2025) used LLMs for translation and synthetic data generation for training data, then performed subject indexing with three Annif-based submodels, Omikuji, Maui-like Lexical Matching (MLLM) and XTransformer. Ranking was performed by the Annif-based XMTC algorithms in the toolkit. The DNB team (Kluge and Kähler, 2025) took an ensemble prompt-based approach to extract keywords from documents by prompting multiple LLMs, and then aggregating the keywords and mapping them to the GND subject headings by a nearest neighbour search. The labels were ranked by employing an LLM to rank the relevance of each label. The La2I2F team (Salfinger et al., 2025) approached the task with vector space matching supported by analogical reasoning and ontology-based retrieval. After mapping the queried document's title and abstract onto the vector space, they are compared to the , finally fused the candidates retrieved as the output. The Homa team (Tekanlou et al., 2025) approached subject indexing as an alignment task. They used OntoAligner, a toolkit for ontology alignment, along with retrieval-augmented generation (RAG) technique for the task.

Our attempt at the task in this second phase by prompting an instruct model directly for the subject indexing task explores the possibilities in integrating external knowledge such as the GND taxonomy into LLMs, handling ambiguity in polysemous terms in different technical contexts and mitigating hallucinations from generative models.

## 2 Task Background

At the second phase of LLMs4Subjects (D'Souza et al., 2025b), two subtasks are released to challenge participants to develop LLM-based ap-

proaches for domain classification and subject indexing, using the dataset of technical records from the TIBKAT open-access catalog, maintained by the Leibniz Information Centre for Science and Technology (TIB) (Deutsche Nationalbibliothek, 2023). Predictions of the subject domains and subject headings should be based on the semantic relationship between the subject, title, and abstract of the record.

Subtask 1 is a multi-label subject domain classification task, where systems are developed to classify a given document into one of more of 28 predefined subject domains according to Subject Classification System LinSearch from the TIB terminology service.

Subtask 2 is a subject indexing task, where systems are expected to tag up to 20 labels for a given document according to the set of over 200,000 subject headings in the GND subjects taxonomy (Deutsche Nationalbibliothek, 2023). The subject tagging capability of systems is expected to align with the annotations in TIBKAT (Technische Informationsbibliothek (TIB), 2023b).

## 3 System Overview

Two systems were developed for the two subtasks, where the system for Subtask 2 utilises predictions from Subtask 1 as a part of its input. For subtask 1, a simple approach of fine-tuning a multi-class classifier is taken to predict LinSearch classes. For subtask 2, we have taken a daring generative approach in attempt to instruct a model to directly generate subject headings despite its hallucinations. Then map the hallucinated pseudo-subjects into the set of labels existing in the GND.

### 3.1 Subtask 1: Multi-Domain Classification of Library Record

A multi-class classifier was fine-tuned with the TIBKAT dataset for the subtask, where titles and abstracts serve as inputs and LinSearch categories as outputs. Since each document can be labelled with multiple LinSearch labels, a threshold on prediction confidence is employed to filter predicted classes in returning final outputs.

The fine-tuned model[1] and training datasets [2] [3]

---

[1] https://huggingface.co/ubffm/xlm_roberta_large_linsearch_classification
[2] https://huggingface.co/datasets/ubffm/linsearch_train_data
[3] https://huggingface.co/datasets/ubffm/linsearch_dev_data

are available on Huggingface.

### 3.2 Subtask 2: Subject Indexing

Since the set of subject headings is too large to be taken as a regular classification task, and there is not enough training data for each subject class, we decided to take a generative approach for this subtask. We have taken a three-step process of 1) classifying with LinSearch domains (Subtask 1), 2) generating GND-like subject headings with metadata by prompting a fine-tuned model, and 3) mapping the pseudo-subjects to the set of existing subjects.

#### 3.2.1 Prompt-embedded dataset

In order to assist the model in learning the semantic relationships and conventional uses of the GND and LinSearch classes, as well as to contextualise the task to be given later at inference, a prompt-embedded dataset with labeled data is created. The dataset does not differentiate the language of the record, which has the advantage of simplicity in structure, as well as the disadvantage that all LLMs used in the pipeline are limited to multilingual pre-trained models. Training data of the instruct model is formatted with the role the agent should act as when performing the task, task description, document title and abstract, LinSearch classes and GND subject IDs and Names. The template format is illustrated in Appendix B. At inference, LinSearch labels predicted from Subtask 1 are integrated into the prompt, sample outputs in JSON are also provided. The template format is illustrated in Appendix C. A slight discrepancy exists between the prompts used during training and those employed at inference, resulting from modifications introduced through subsequent prompt experimentation. These adjustments were intended to enhance output quality and minimize formatting errors.

#### 3.2.2 Generating Pseudo-classes

We employ a Llama instruct model fine-tuned with the prompt-embedded dataset and prompt it directly with a subject indexing task to generate the foundation of the subject headings. Fine-tuning is performed to enable the model to learn the patterns and conventions associated with subject indexing using the GND and specifically the TIBKAT dataset. While the model might have learnt the implicit relations between texts and some phrases that are repetitively used as subject headings, it does not have explicit knowledge of the set of predefined

labels in the GND dataset and is prone to hallucination. We thus address this limitation by prompting the model to produce supplementary contextual information.

The model is given a document with a title, an abstract and a list of LinSearch labels predicted from Subtask 1, and prompted to label it with exactly 20 GND subjects in the sample JSON format. It is asked to provide not only the subject index name but also associated metadata, such as a unique "Code" (e.g., gnd:XXXXXXX-X), "Alternate Name" (synonyms), and "Related Subjects" (connected topics). An example of the sample output format is displayed in Appendix D. This structure replicates the GND dataset format, maintaining consistency with professional cataloging standards. This additional information enables the post-processing step, where hallucinated or inconsistent classes can be mapped back to the correct GND labels by taking the extra metadata as context, ensuring alignment with the structured dataset.

### 3.2.3 Fallback: Keywords Extraction

A significant limitation of relying on generative models to perform complex tasks is the increased unpredictability of outputs. Empirical observations demonstrate instances where the generated text fails to be parsed as a valid JSON object, speculated to be due to two primary factors. Firstly, the generated output may exceed the token limit, causing truncation. Secondly, the model may drift into generating repetitive content over extended sequences, reducing output utility.

Therefore, in instances where the model output cannot be extracted as a JSON object after simple postprocessing, a fallback strategy based on keyword extraction is employed to ensure essential information is retained. Specifically, KeyBERT (Grootendorst, 2020) is used to extract up to 10 keywords, each comprising 1-3 tokens. These keywords are derived from the concatenated text of the title and abstract, enabling downstream processing despite structural limitations in the model's output. This approach serves as a robust mechanism to handle edge cases while maintaining the utility of the extracted information for subsequent computational tasks.

### 3.2.4 Mapping to GND Subject Headings

The hallucinated pseudo-classes are mapped to subject headings in the GND set by comparing their respective encodings. By encoding the subject head-

ings with the classes they are relevant to, the extra context given provides the model with the technical context each subject belongs to, to assist disambiguation for polysemous terms. Prior to mapping, the "Name," "Alternate Name," and "Related Subjects" fields of each GND subject are concatenated and encoded using a sentence-BERT model. Similarly, for each model-generated pseudo-class, the fields "Name," "Alternate Name," and "Related Subjects" are extracted, concatenated, and encoded using the same sentence-BERT model. The resulting embeddings from the pseudo-classes are then compared with the GND set encodings to establish the mapping. This approach utilises the creative output of the generative model, while ensuring the final outputs are constrained to align with the pre-defined set.

While the full pipeline remains unpublished, the fine-tuned instruct model is accessible on Huggingface[4]. Training datasets[5][6] are also publicly accessible.

## 4 Experimental Setup

### 4.1 Subtask 1

A dataset is created by extracting the document title, abstract and LinSearch labels of each document in the TIBKAT dataset. It is further populated by splitting entires with more than one LinSearch tags into multiple entries. The resulting training and validation set have approximately 78,900 and 13,200 entries, where each of the entries has one LinSearch label.

The large XLM-RoBERTa model (Conneau et al., 2020) is chosen for the task for its balance between performance and model size. With 561M parameters, it delivers competitive multilingual classification accuracy without excessive computational cost. This makes it a practical and powerful choice across a wide range of real-world multilingual NLP tasks, especially with a lower complexity. The model is fine-tuned as a classifier with the Huggingface trainer module. It processes input text strings and returns a list of predicted LinSearch class with their corresponding confidence scores. In order to return multiple classes in the final output, a threshold of 0.05 in confidence score is applied to filter

---

[4] https://huggingface.co/ubffm/llama-3. 2-3b-instruct-unsloth-bnb-4bit-gnd-subjects

[5] https://huggingface.co/datasets/ubffm/prompt_ gnd_with_linsearch_train_data

[6] https://huggingface.co/datasets/ubffm/prompt_ gnd_with_linsearch_dev_data

out unlikely predictions. The remaining classes are then ranked in descending order based on their confidence scores. Considering that the confidence scores of all classes add up to 1, the threshold is set so that 1 to 3 classes will be returned in most cases, to better resemble the number of LinSearch classes labeled in the majority of TIB records. Appendix A presents some sample predictions and scores to demonstrate the usefulness of the threshold.

## 4.2   Subtask 2

A dataset is created as in Subtask 1, then transform into a prompt-embedded task description, as described in 3.2.1. The dataset is then used to fine-tune a Llama 3.2 3B instruct model (Meta, 2024) with Unsloth (Unsloth, 2025) to include compute-efficient measures. The model is trained using Unsloth with 4-bit quantisation and rank-stabilized LoRA. As a result, only 32.5% of all trainable parameters, which is 982,515,712 parameters, are updated during training. Although GPU access at a high-performance data center was granted at a later stage by collaboration, initially the project was developed with one RTX 4080 GPU at our local institute. Therefore, we decided to stick to the decision with using Unsloth and a quantised model as we initially started, with the limited access to GPUs in most academic libraries in mind, as well as to align with the shared task's highlight on developing energy- and compute-efficient LLM systems in this phase.

The fine-tuned model is prompted with a subject assignment task, as described in 3.2.2 with a maximum token count of 2000 tokens, to obtain GND-like subject classes with extra information on top of the names of the classes. When the returned text is truncated due to the token limit, a simple postprocessing step is applied to cut off the output at the last complete class for which the full set of metadata can be extracted. When no information at all from the generated text can be extracted, mostly due to malformatted JSON text, a fallback measure is employed to extract keywords from the title and abstract, as described earlier in 3.2.3.

Since there is no guarantee that exactly 20 classes will be generated from the model, another postprocessing step ensures that there will be 20 classes at the output. Where $n \cdot p$ is the number of pseudo-classes extracted and $l$ is the number of labels each pseudo-class would be mapped to, $n \cdot p \cdot l > 20$ is calculated for the minimum number of labels needed from each generated pseudo-

|        | Precision       | Recall          | F1              |
|--------|-----------------|-----------------|-----------------|
|        | macro / micro   | macro / micro   | macro / micro   |
| Overall | 0.65 / 0.66    | 0.65 / 0.66     | 0.65 / 0.66     |
| Article | 0.72 / 0.73    | 0.72 / 0.73     | 0.72 / 0.73     |
| Book   | 0.65 / 0.66     | 0.65 / 0.66     | 0.65 / 0.66     |
| Confer. | 0.65 / 0.66    | 0.65 / 0.66     | 0.65 / 0.66     |
| Report | 0.62 / 0.63     | 0.62 / 0.63     | 0.62 / 0.63     |
| Thesis | 0.67 / 0.68     | 0.67 / 0.68     | 0.67 / 0.68     |

Table 1: Subtask 1 quantitative evaluation results by record type

class. Subsequently, $l$ GND subject headings are extracted from each pseudo-class by comparing the $n$ most similar subject labels with cosine similarity, as described in 3.2.4. In the output, ranking of the classes follows the order they are generated from the model. For example, when 3 pseudo-classes in the order of $p_1$, $p_2$ and $p_3$ are extracted, $l$ will be 7. Seven labels mapped from $p_1$ will be placed at the top of the list, followed by 7 labels from $p_2$, then 7 from $p_3$. When the number of labels returned are over 20, the final output will be cut off at 20.

## 5   Results and Limitations

### 5.1   Subtask 1

Performance in Subtask 1 is overall at 65% for precision, recall and F1, which placed us at 1$^{st}$ place due to low participation, but in the greater picture it is not particularly impressive as a classifier.

Table 1 presents the quantitative evaluation results for Subtask 1, broken down by record type. The model achieves consistent performance across all metrics, with overall macro and micro scores for precision, recall, and F1 hovering around 0.65–0.66. Among the individual categories, articles yield the highest performance, with an F1 score of 0.73, indicating that the model is especially effective at classifying this record type. Theses and books also show relatively strong performance, with F1 scores of 0.67 and 0.66, respectively. On the other hand, reports are the most challenging category, receiving the lowest F1 score (0.63), suggesting some ambiguity or variability in this class's metadata features. The small difference between macro and micro scores further indicates balanced performance across classes, with no severe class imbalance or overfitting to dominant types.

### 5.2   Subtask 2

Figure 1 presents the performance of the subject indexing system across different record types and

cut-off values $K \in \{5, 10, 15, 20\}$. Overall, the system shows a linear and gradual improvement in ranking metrics as $K$ increases. For instance, overall NDCG improves from 0.1193 at $K@5$ to 0.1465 at $K@20$, and recall increases correspondingly from 0.1282 to 0.2003.

Performance varies significantly by record type. *Book*, with the most training data, consistently outperform other categories across all metrics, achieving the highest NDCG (0.1549) and recall (0.2145) at $K@20$. *Conference* and *Report* records also demonstrate competitive performance, with F1 scores comparable to that of books. In contrast, *Articles* show the weakest performance, particularly in precision (e.g., 0.0138 at $K@20$) and F1 (0.024), suggesting difficulty in identifying relevant GND subjects for this type. *Thesis* fall in the mid-range, with moderate gains across $K$ levels but overall lower recall and F1 compared to books and reports.

The results highlight both the general scalability of the system across increasing $K$ values and the variability in performance across document types, likely due to differences in metadata richness, language structure, and topical consistency.

At test output generation, approximately one third of the cases triggered the fallback measure and their outputs were replaced by keywords. We speculate that this behavior is primarily due to the model's limited capacity, as it is a relatively small 3B parameter model quantised to 4 bits, which may contribute to instability in its outputs. There were attempts to adjust the temperature and repetition penalty to reduce invalid JSON formats. However, as both parameters severely interfere with the creativity of the generative model, we decided to settle on the trade-off between creative outputs and formatting correctness at the final model. Another challenge arising from the model's instability is the generation of unusually short or long outputs (e.g., fewer than 5 words or more than 50), particularly during phases of repetitive output. These length deviations lead to inconsistencies when such outputs are compared to embeddings derived from texts with typical lengths (10–20 words), which negatively impacts the reliability of embedding-based similarity matching.

## 6   Conclusion and Proposed Extensions

We present two systems for bilingual subject classification and indexing using fine-tuned LLMs and

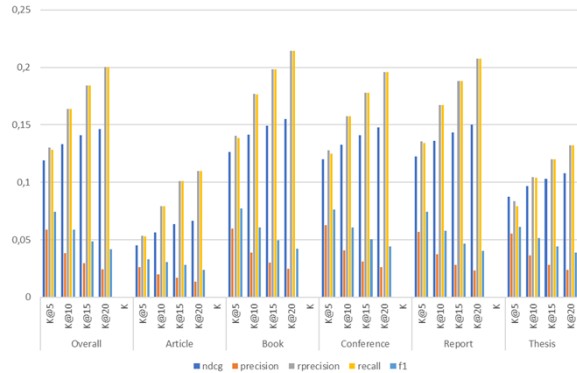

Figure 1: Subtask 2 quantitative evaluation results by record type

structured knowledge integration. The system developed for subtask 1 achieved consistent F1 scores above 0.65 across domains and was placed at the first place. For Subtask 2, an innovative approach utilising model hallucinated outputs mapped to GND labels was proposed. Although results were not comparable to the two best teams, we have attempted to push the limits of prompting to directly tackle the subject indexing task as is, and to address challenges such as polysemy, hallucination, and context ambiguity.

To further improve the performance of the subject indexing pipeline, we propose 3 potential extensions for the second subtask based on the current structure. Firstly, JSON could be opted out for other machine-readable formats to reduce output formatting issues, thus maximise the utility of the generative model. Secondly, implementing a more sophisticated fallback mechanism or integrating an alternative tool could enhance the alignment between extracted keywords and the structure of GND subject classes, thereby improving the accuracy of embedding-based similarity matching. Thirdly and similarly, post-processing generated texts from the instruct model to constrain the length of text used for encoding pseudo-classes may also facilitate more effective mapping to GND subjects.

## Acknowledgments

This work was partially funded by the DFG under FID Linguistik (326024153). We gratefully acknowledge support from the hessian.AI Service Center (funded by the Federal Ministry of Education and Research, BMBF, grant no. 01IS22091) and the hessian.AI Innovation Lab (funded by the Hessian Ministry for Digital Strategy and Innovation, grant no. S-DIW04/0013/003).

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

## A   Sample LinSearch classification results

| Record ID | Gold label | Prediction | Score (Pred) |
|---|---|---|---|
| 3A371288258 | mat | mat | 0.99773 |
| | | phy | 0.00053 |
| | | oek | 0.00035 |
| | | elt | 0.00022 |
| | | bio | 0.00021 |
| 3A46120360X | lit | sow | 0.78562 |
| | lin | phi | 0.14748 |
| | | his | 0.05145 |
| | | lit | 0.00785 |
| | | oek | 0.00310 |
| 3A873554329 | sow | oek | 0.38645 |
| | oek | sow | 0.36542 |
| | pae | pae | 0.24364 |
| | | tec | 0.00201 |
| | | phi | 0.00077 |

Table 2: Top 5 LinSearch classification results for 3 records in the development set

## B   Template of prompt-embedded training data

```
### Act as:
Act as an expert librarian, who is
familiar with the GND (the integrated
authority files) and text processing.

### Task:
You are now tasked to assign subject titles
to documents according to the database of
GND subject titles. Given the document
title and abstract, assign the 10 most
suitable GND subject titles for the
document with no repetition.
Annotate GND Subjects in JSON format based
on document title, abstract and its class
in the subject classification system
LinSearch. You have been trained on a
dataset of GND and LinSearch annotated
documents.
Use your knowledge to match new documents
with GND subject data as closely as
possible to the style in the training
dataset. For each of the GND subject titles,
provide the name of the subject titles,
its names and related subjects.
```

Make sure all the information you provide are as listed on the GND database and rank your results by the likelihood they would be assigned by an expert librarian.

### Title:
{title}

### Abstract:
{abstract}

### GND ID:
{list of GND codes}

### GND Class:
{list of GND classes}

### LIN Search class:
{list of LinSearch Classes}

## C  Template of prompt at inference

You are a librarian and an expert in the GND (the integrated authority files):
{truncated GND subject headings in dataframe}

You also have the knowledge of LIN Search classification:
- arc Architecture
- bau Civil engineering
- ber Mining
... (content omitted)

### Given the following details:
- Document Title: {title}
- Document Abstract: {abstract}
- Document LIN Search class:
{list of predicted LinSearch classes}

### Your task:
Using your expert knowledge in the conventional use of the GND, you are now tasked to assign **exactly 20** subject titles to documents according to the database of GND subject titles. Your task is to annotate documents with GND subjects in JSON format based on their content (title and abstract). Use your knowledge of GND annotations and follow the established patterns from past training data.

1. Analyze the title, abstract and LIN Search class of a document and infer GND subject annotations.
2. Produce the output in structured JSON format, maintaining consistency with examples provided.

Rules:
- Always extract subjects from the core concepts mentioned in the title/abstract.
- Include synonyms, variant names, and related subjects in arrays.
- Follow the structure exactly as shown in the examples.
- Always quote the JSON format output between ```json and ```.
- No more than 20 alternate names is allowed for each subject.
- No more than 20 related subjects is allowed for each subject.

### Instructions:
1. Analyze the provided document to identify its core themes, topics, and concepts.
2. Select the most appropriate **20 unique GND subject titles** that align with the content, in the same style as the dataset you were trained on.
3. Rank the GND subject titles according to the relevance to the document, and the likelihood they would be assigned by an expert librarian.
4. Format the output as JSON, adhering to the structure provided below. For each of the subject titles, reply with the name of the subject titles, its names and related subjects in a list.
5. Make sure to complete the JSON output before you hit the maximum token limit, to ensure the correctness of the JSON format.

### Output example:
```json
{
"document_title": "Energie- und Lademanagement für eine CO2-neutralen Beladung von batterieelektrisch betriebenen Service-Fahrzeugen auf dem Flughafenvorfeld",
"assigned_subject_titles": [
{
"Code": "gnd-4014736-8",
```

```json
    "Name": "Energieversorgung",
    "Alternate Name": ["Energie",
"Energieversorgungssystem", "Energiesystem"],
    "Related Subjects": ["Versorgung",
"Energieerzeugung", "Energiewirtschaft"]
    },
    {
    "Code": "gnd-4068598-6",
    "Name": "Erneuerbare Energien",
    "Alternate Name": ["Sanfte Energie",
"Erneuerbare Energiequelle", "Regenerative
Energiequelle", "Regenerative Energie",
"Erneuerbare Energie", "Alternativenergie",
"Alternativenergien", "Regenerative Energien",
"Alternative Energiequelle", "Grüne Energie",
"Green energy"],
    "Related Subjects": ["Energiequelle",
"Erneuerbare Ressourcen"]
    },
    ...
    ]
}
```

```
        "Related Subjects":
["Konditionalisierung",
"Unsicherheit", "Vorkalkulation",
"Prioritäten", "Schatzungen"]
    },
    ... (content omitted)
    ]
}
```

## D  Output example at model inference (partial content omitted)

```json
{
  "document_title": "Probability matching
  priors : higher order asymptotics",
  "assigned_subject_titles": [
    {
      "Code": "gnd-4000000-9",
      "Name": "Statistik",
      "Alternate Name": ["Statistics",
      "Statisztika", "Statistiche",
      "Statistique", "Estimation",
      "Analyse statistique"],
      "Related Subjects": ["Inferenz",
    "Schätzung", "Methodik der Statistik",
      "Theorie der Wahrscheinlichkeit",
      "Wahrscheinlichkeitsrechnung"]
    },
    {
      "Code": "gnd-4021000-7",
      "Name": "Bayesische Statistik",
    "Alternate Name": ["Bayesian statistics",
      "Bayésienne statistique",
      "Bayes'che Methode",
      "Méthodologie bayésienne",
      "Théorie des Bayes"],
```
