# OpenReview forum: "UBFFM at the GermEval-2025 LLMs4Subjects Task:  What if we take "You are an expert in subject indexing" seriously?"
_GSCL.org/KONVENS/2025/Workshop/GermEval — GermEval25 Oral_

### Official Review · Reviewer_E1DH · 2025-08-09
**Proper citations, clarification of figures, simplification of complex sentences, detailed explanations of model choices and fine-tuning, a concise task summary, and discussion of dataset biases.**

**Rating:** 3
**Confidence:** 5

**Review:**

The paper presents an innovative approach to subject indexing with some notable contributions. The method uses generated pseudo-subjects combined with extra information is creative and addresses the challenge of large label spaces effectively. The practical consideration of computational constraints and the comprehensive fallback mechanism for handling edge cases show thoughtful system design. The multi-step pipeline that integrates domain classification with subject indexing is well-motivated. The paper title and page limit is taken into consideration.


Still, I can recommend some improvements:

Strongly recommended:
1. Cite Unsloth properly at the section 4.2
2. Fix figure 1 caption: it currently refers to subtask 1, but the figure actually illustrates subtask 2.

Recommended:
1. Some sentences, especially in sections 3.2.1 and 3.2.4, are quite long and may be difficult to follow. Please break them into shorter, simpler sentences to improve clarity.
2. I kindly recommend to include these information:
    - Explain the specific reasons for choosing XLM-RoBERTa as the base model for classification and Llama 3.2 as the base model for generative tasks.
    - Provide more details on the fine-tuning strategy beyond just mentioning the packages used. For example, specify if the fine-tuning involved contrastive learning or another approach.
    - Include fine-tuning configurations and hyperparameters.
3. There is no need to describe the task in detail in section 2 Task Background. A brief summary with a citation of the task’s original paper is enough.
4. Please discuss potential dataset biases in TIBKAT. Are there any domain-specific or language-specific biases that might affect the generalizability of your approach?

Some personal opinion about the work:
- The percentage of edge cases and invalid JSON outputs during the Generative Pseudo-classes phase is not addressed. Analyzing the quality of these outputs would help in evaluating whether improvements are possible using other base models or fine-tuning strategies.

- You applied a very low confidence threshold (0.05) in subtask 1. It would be informative to know how many labels on average exceeded this threshold per sample. If the threshold is so low, it may indicate that the model was not well trained or that the confidence scores are skewed toward certain values. Please clarify this point.

**Summary:**

The paper presents a new method for subject indexing that effectively tackles the challenges of large label spaces by using generated pseudo-subjects and additional information. It features a well-designed multi-step pipeline that combines domain classification with subject indexing while considering computational constraints.

---

### Official Review · Reviewer_fgBt · 2025-08-14
**This was a reasonable attempt at LLM-based subject indexing, but the results were not particularly good and some of the methods were implemented in a questionable way.**

**Rating:** 3
**Confidence:** 5

**Review:**

Strengths:

- The system addressed both subtasks (unlike other participating systems)
- Consistent classification results for subtask 1 across different record types
- Relatively lightweight LLM-based approach, using a small (3B) fine-tuned language model, fitting well to the theme of compute efficient LLM-based subject indexing
- I think it's a good idea to prompt the LLM to produce not just individual GND labels, but a richer representation of a (possible) GND subject with alternate names and related subjects. This could make it more likely that suitable matches in GND are found via BERT embeddings.
- The paper was relatively clear and provided a reasonable amount of detail considering the strict page limit.

Weaknesses:

- The paper doesn't discuss the multilingual aspects of the task at all, namely, that some records were in German, others in English, or a mixture of both. I got the impression that all records regardless of language were simply pooled together, but this could be stated more directly.
- The fine-tuning of the LLM, by creating the "prompt-embedded dataset", is performed in an unconventional way that I think is not ideal for the purpose of teaching the task to the LLM. The more conventional method would be to use the same prompt template both for fine-tuning and eventual inference, with the fine-tuning dataset also including the ideal response that includes the gold standard LinSearch classes and GND subjects. Thus the LLM would be nudged already during fine-tuning towards producing such responses.
- There is no comparison to baseline methods for either subtask 1 or 2, nor any testing of different methods or variations.
- It is unclear whether the fine-tuning of the LLM actually improved results compared to just carefully prompting the original model. The fine-tuning is both labor and compute intensive, so doing it should be justified.
- The results for subtask 2 weren't particularly good compared to other systems/teams, and there was no error analysis and/or ideas about potential improvements and future work.
- This was submitted as an anonymous work, which I think was not advised for GermEval (?). There were no links to code (e.g. on GitHub) or the fine-tuned models (e.g. on HuggingFace)


Questions to the authors
- Why did you prompt the LLM to include the document_title and the GND IDs of the subjects in the response? I don't see any use for this information, as the title was already known and given in the prompt, and the GND IDs were, in my understanding, not used; instead, the LLM-generated (or "hallucinated") GND subjects were mapped to actual GND subjects using the BERT model based on other information such as labels, but not the ID/code.
- You mention that you had to use a fallback model (KeyBERT) in cases where the LLM fails to produce consistent JSON output. How often was this necessary for generating test set predictions? Did you try other strategies to prevent invalid JSON output, for example increasing the limit of generated tokens or adjusting the temperature and/or repetition penalty (which often helps avoid repetitive output)?

Possible mistakes, typos etc:

- lines 21-22: paper states that LinSearch is developed by DNB; I think it's developed at TIB
- line 32: could be a matter of opinion / taste, but I wouldn’t necessarily use the word recent in conjunction with applying machine learning for automated indexing (Annif prototype was released 8 yrs ago), LLMs are another beast of course
- lines 43-44: "Ranking was performed by using two XMTC metrics…"; I believe this mixes up ranking (of results, i.e. suggested GND subjects) with evaluation of candidate models. Ranking of GND subjects was performed internally by the Annif-based XMTC algorithms (here called sub-models).
- line 50: "neighbout" -> "neighbour" or "neighbor"
- lines 71-72: the DNB citation is puzzling; I think some other source such as the TIB or TIBKAT web site would be more appropriate
- line 239: Unsloth citation missing
- line 286: "cut of" -> "cut off"
- Figure 1 caption: it says Subtask 1 but actually represents Subtask 2
- Maybe this is a matter of style and opinion, but I find statements like "Two systems are developed" (line 91), "A multi-class classifier is fine-tuned" (103) and "A dataset is created" (216) a bit unclear. Switching to past time ("were developed", "was fine-tuned", "was created") would be better, and even better would be changing to active voice ("We created a dataset").

**Summary:**

The paper describes a system for automated subject indexing that targets both subtasks of the LLMs4Subjects shared task. For the first subtask, a XLM-RoBERTa based classifier is fine-tuned using the provided training data. For the second subtask, a LLM (Llama 3.2 3B Instruct) is fine-tuned on prompts derived from the training data set and then applied on the test set records. Predicted subject labels are mapped to GND subject labels using sentence-BERT embedding similarity. Both systems were evaluated by the shared task organizers. For subtask 1, this was the only participating system and was thus ranked 1st. For subtask 2, the system was ranked 3rd out of three systems, with scores much below the others.

---

### Decision · Program_Chairs · 2025-08-14

Accept (Oral)